# Tumor-Promoting Activity and Proteomic Profiling of Cisplatin/Oxaliplatin-Derived DAMPs in Cholangiocarcinoma Cells

**DOI:** 10.3390/ijms231810540

**Published:** 2022-09-11

**Authors:** Worawat Songjang, Chatchai Nensat, Nitirut Nernpermpisooth, Porrnthanate Seenak, Panyupa Pankhong, Noppadon Jumroon, Sarawut Kumphune, Arunya Jiraviriyakul

**Affiliations:** 1Department of Medical Technology, Faculty of Allied Health Sciences, Naresuan University, Phitsanulok 65000, Thailand; 2Integrative Biomedical Research Unit (IBRU), Faculty of Allied Health Sciences, Naresuan University, Phitsanulok 65000, Thailand; 3Chulabhorn International College of Medicine, Thammasat University, Pathum Thani 12120, Thailand; 4Department of Cardio-Thoracic Technology, Faculty of Allied Health Sciences, Naresuan University, Phitsanulok 65000, Thailand; 5Biomedical Engineering Institute (BMEI), Chiang Mai University, Chiang Mai 50200, Thailand

**Keywords:** DAMPs, cholangiocarcinoma, HSP, S100, HMGB1, proteomic profiling, tumor-promotion

## Abstract

Damage-associated molecular patterns (DAMPs) are well recognized as the molecular signature of immunogenic cell death (ICD). The efficacy of drug-induced ICD function may be impacted by the precise ratio between immunostimulatory and immunoinhibitory DAMPs. Tumor-derived DAMPs can activate tumor-expressed TLRs for the promotion of tumor cell motility, invasion, metastatic spread and resistance to chemotherapeutic treatment. Herein, drug-induced DAMPs’ expression and their role in tumor progression are utilized as one crucial point of evaluation regarding chemotherapeutic treatment efficacy in our study. Cisplatin and oxaliplatin, the conventional anticancer chemotherapy drugs, are emphasized as a cause of well-known DAMPs’ release from cholangiocarcinoma (CCA) cells (e.g., HSP family, S100, CRT and HMGB1), whereby they trigger Akt, ERK and Cyclin-D1 to promote tumor activities. These findings strengthen the evidence that DAMPs are not only involved in immunomodulation but also in tumor promotion. Therefore, DAMP molecules should be considered as either targets of cancer treatment or biomarkers to evaluate treatment efficacy and tumor recurrence.

## 1. Introduction

CCA is a malignant type of tumor that begins as the epithelial cell lining in the bile ducts and can migrate to other areas of the liver. Overall, 15% of all primary liver tumors are represented as CCA, and its incidence is increasing worldwide. Regarding CCA treatment, depending on the anatomical subtype of the disease, surgical procedures, locoregional therapy, cytotoxic therapy, targeted therapy and immunotherapy are among the treatment options [1].

Whether chemotherapy aims to induce apoptosis or necrosis directly, there is growing evidence to support the idea that chemotherapy may stimulate damage-associated molecular pattern molecules (DAMPs) of tumor cells [2,3,4,5,6]. While many DAMPs are secreted, others collect on the outer surface of the plasma membrane or are exposed de novo. In the last stages of the cell death process, other DAMPs are also generated as breakdown products. The non-immunological role of these molecules has been mainly suggested prior to release or exposure to the extracellular surface. Immunogenic cell death (ICD) is defined by DAMPs’ release or surface expression, which are related to the long-term efficacy of anti-cancer therapies by combining the direct elimination of cancer cells with antitumor immunity [7]. Although the synthesis of necessary ICD for anti-cancer mechanisms has been proven, there is compelling evidence that DAMPs are involved in a tumor-promoting mechanism [8,9,10]. Several DAMPs, such as HMGB1, heat shock proteins, ATP, CRT and S100 proteins, have been reported as promoting tumor progression via PAMP activation and advanced glycation end-product (RAGE) receptors, including the activation of immunosuppressive cells [11]. Notably, HMGB1 was found to promote tumor proliferation and migration via activation of the RAGE receptor, Wnt/β-Catenin and MAPK pathways [8]. Interestingly, S100 and HSP are involved in the induction of tumor progression and are also important therapeutic targets for cancer treatment [12,13].

Herein, we focus on the distinct role of DAMPs in the tumor promotion mechanism. We firstly describe the proteomic profiling of ICD-associated DAMPs generated by routinely used chemotherapeutic drugs in CCA medical care. The relation of these DAMPs to tumor-promoting activity is also discussed. The key contribution of this study is to emphasize chemotherapy-derived DAMPs in cancer, which could lead to the development of novel treatments or biomarkers.

## 2. Results

### 2.1. Cisplatin and Oxaliplatin Caused CCA Cell Toxicity

Induction of DAMPs’ release is involved in many mechanisms of cell injury. We demonstrated that chemotherapeutic agents cisplatin and oxaliplatin are the cause of KKU213L5 cell death, as shown in Figure 1A. Therefore, cisplatin and oxaliplatin are representatives of chemo drugs for the induction of DAMPs in CCA cells, observed by using the IC50 at 24 h, which are 364.2 and 348.7 µM, respectively. At this concentration, they significantly decreased the cell viability of KKU213L5 and significantly promoted cell apoptosis via increased annexin V expression, increased active-caspase 3 expression and markedly decreased anti-apoptosis markers, as shown in Figure 1B,D. Moreover, these drugs increased LDH (lactate dehydrogenase) activity when compared with untreated groups (Figure 1C). This result indicated that cisplatin and oxaliplatin exhibit a cytotoxic effect on CCA cells.

### 2.2. Cisplatin and Oxaliplatin Induced DAMP Expression in CCA Cells

DAMP profiling of chemo-drug-derived conditioned medium was performed via proteomic analysis using label-free nano-LC-MS/MS. Samples were digested and sequentially analyzed for relative intensity by using MaxQuant. A total of 1227 proteins were detected in the control and cisplatin and oxaliplatin treatment groups. We found that chemo drugs proficiently induced the overexpression of 45 DAMPs, such as the heat shock protein family, S100 proteins, HMGB2, calreticulin, annexin A2-3 proteins and mitochondria proteins (Table 1). During cisplatin treatment, 36 DAMPs were increased more than 2.5-fold compared with the controls, and the predominant DAMPs were histone, ATP synthase and galectin-8 proteins. For oxaliplatin treatment, 34 DAMPs were elevated more than 2.5-fold, with a predominance of annexin A2 and 10 kDa heat shock protein, as shown in Figure 2.

To confirm the remarkable effect of chemo drugs in inducing DAMP release, treatment-conditioned medium was measured, representing DAMPs including HMGB1, HSP90, S100A12 and extracellular ATP, with the use of other biological assays. Cisplatin treatments caused HMGB1, HSP90, S100A12 and ATP secretion due to 2–4-fold increments when compared with the control group. In the same direction, DAMP release was induced by oxaliplatin, with the exception of HSP90 (Figure 3A–D). Moreover, chemo-drug-treated cells were stained with anti-CRT antibodies before being quantified using a fluorescent microscope and flow cytometer. The results indicated that cisplatin and oxaliplatin induced the overexpression of CRT via expanded cytoplasm localization (Figure 3E). Moreover, cell membrane staining showed highly expressed CRT on chemo-drug-treated cancer cells, as shown in Figure 3F. Taken together, these observations confirmed that cisplatin and oxaliplatin treatment induced DAMP expression and production.

### 2.3. Chemo-Drug-Derived DAMPs Promoted CCA Cells’ Proliferation and Migration

To evaluate the biological activity of chemo-drug-derived DAMPs, a conditioned medium of cisplatin- and oxaliplatin-treated CCA cells was inspected in terms of in vitro cancer proliferation and migration. Interestingly, they significantly elevated cell proliferation when compared with the control medium, especially at 48 and 72 h of incubation (Figure 4A). As well as proliferation, cisplatin- and oxaliplatin-treated conditioned medium exhibited an increment in migration ability via the wound scratch assay, as shown in Figure 4B. These results indicated that chemo-drug-derived DAMPs enhanced cancer proliferation and migration.

### 2.4. Chemo-Drug-Derived DAMPs Activated the Pro-Survival Signal of CCA Cells

Survival signaling proteins were further investigated to verify the protumor enhancement of DAMP-conditioned medium. Increasing incubation time caused the overexpression of pAkt and pERK, which are important representatives of mitogenic signaling pathways. In addition, chemo-drug-derived DAMPs elevated the expression of cyclin D when compared with the untreated group; this was observed in both KKU213-L5 and KKU100 cells (Figure 5A). To confirm the role of DAMPs in cancer cell proliferation, the cells were treated with chemo-drug-derived conditioned medium with or without PI3K and ERK inhibitors. The results showed that the additional inhibitor attenuated the cell proliferation effect (Figure 5B). To confirm the role of chemo-drug-derived DAMPs in protumor activities, protein–protein interaction analysis was used to analyze the possible contribution of overexpressed DAMPs and cancer pro-survival proteins. The results demonstrated that 15 DAMPs were involved in Akt activation, such as annexin A1-2, CRT, HMGB1, the HSP family and galectin. Moreover, the activation of MAKP signaling is associated with eight types of DAMPs, such as S100, peroxiredoxin, the HSP family and HMGB1. Notably, some of them are involved in cell-cycle-regulated protein expression, including the HSP family, galectin and histone protein (shown as Figure 6). Taken together, these results confirm that cisplatin- and oxaliplatin-derived DAMPs are involved in protumor activities via pro-survival protein activation.

## 3. Discussion

Chemotherapy is virtually the standard cancer treatment, either in adjuvant or palliative care. Treatment of CCA patients with first-line systemic chemotherapy standard significantly prolongs overall survival (OS) for patients treated with gemcitabine and cisplatin, compared with those treated with gemcitabine alone (11.7 months vs. 8.1 months, *p* < 0.001) [14]. Moreover, the combination of gemcitabine and oxaliplatin exhibits a modest antitumor response, indicating that oxaliplatin can be applied instead of cisplatin [15]. Previous studies showed that cisplatin and oxaliplatin promoted DAMPs’ production, including HMGB1, HSP90/70 and CRT, which play a role in the activation of the antitumor response, known as ICD [16,17]. However, the contradictory function of DAMPs is worthy of attention. Downstream signaling of RAGE and TLR activation by DAMPs, including MAPK and Wnt/β-Catenin signaling, is associated with protumor activities [8]. In this study, we first performed DAMP profiling and assessed their role in protumor activities using cisplatin- and oxaliplatin-derived DAMPs.

DAMP expression following cisplatin and oxaliplatin treatment was demonstrated, such as HMGB1, the HSP family, CRT, S100, galectin and ATP, which are related to cell death induction. Cell apoptosis is the cause of intracellular and nuclear substances’ externalization on the cell membrane, which mostly mediates the ER stress mechanism [18]. The nuclear molecules, such as HMGB1, histones, exRNAs and cfDNA, as well as ATP, have been shown to be released by apoptosis [19,20,21]. In another direction, DAMP release has been reported to have an association with the necrosis mechanism. Passive release from plasma membrane rupture is a characteristic mechanism for HMGB1, ATP, histones, HSPs and other cytoplasm substances’ release from dead cells [18,22,23]. Although, in this study, we only observed cell apoptosis, this could naturally progress to secondary necrosis [24]. Therefore, cisplatin- and oxaliplatin-derived DAMPs on CCA cells could be released from either apoptosis or necrosis.

Forty-five overexpressed DAMPs were confirmed by proteomic and bioassay analysis. The highly abundant DAMPs, including the HSP family, annexins, galectin, HMGB1, peroxiredoxin, CRT and S100 proteins, were significantly released in the conditioned medium after cisplatin and oxaliplatin treatment. Moreover, we observed ATP release, as measured by the fluorometric assay, related to the upregulation of ATP-synthase protein by proteomic analysis. Despite the notable role of DAMPs involved with ICD and the activation of the antitumor immune response, upregulation of DAMPs is associated with poor prognosis in cancer patients. For example, HMGB1 overexpression was correlated with poorer OS in several cancers when detected by immunohistochemistry in tissues and enzyme-linked immunosorbent assay in serum [25]. The overexpression of HSPs is associated with tumor progression [26] and poor prognosis in gastric cancer and colorectal cancer patients [27]. Interestingly, treatment with cisplatin- and oxaliplatin-derived DAMPs promoted CCA cells’ proliferation and migration (Figure 3). In another direction, whole blood ex vivo cytokine production was measured upon challenge with these DAMPs. The results showed that DAMP-conditioned medium contributed to the production of pro-inflammatory cytokines, such as TNF-α and IL-6 (manuscript in preparation). These results strengthen the dual and contradictory roles of DAMPs in CCA, which can be observed during conventional chemotherapy treatment.

Alteration of signaling pathways that control cell growth and division, cell death, cell fate and cell motility is a factor in tumorigenesis and cancer progression. The Ras-ERK and PI3K-Akt pathways are important regulators of normal cell proliferation, and signal alteration is an example of an oncogenic signaling pathway [28]. We demonstrated that the cisplatin- and oxaliplatin-derived DAMPs activated the phosphorylation of ERK and Akt. There is a range of evidence indicating that the target receptors of DAMPs are TLRs and RAGE receptors [11], which belong to many downstream signaling pathways, such as pro-inflammatory regulation through NF-κB signaling, cell survival and migration regulation through the MAPK pathway [29,30,31]. It has been mentioned that chemotherapeutics treatment also induced exosomal DAMPs production that could activate target cell thought endocytosis pathway [32]. Therefore, we postulated that DAMPs activated cancer protumor activities either cell receptor or intracellular receptor activation by phagocytosis mechanism. Our analysis of protein–protein interactions showed that over 15 and 8 DAMP molecules are involved in Akt and ERK signaling (Figure 6). Although these results indicate that chemotherapeutic-derived DAMPs promoted CCA proliferation and migration, they mediated pro-survival pathway activation. In another direction, cell death releases other proteins that may impact protumor activity, such as growth factors, signaling proteins and cytokines, as shown in Appendix A. Some of them are involved in Akt and ERK activation (Appendix A). Herein, in the real clinical setting, DAMPs do not only play a role in protumor activation, but some other proteins also contribute to this bioactivity. For further study, the bioactivity of each DAMP should be investigated for its obvious mechanism and further considered as a therapeutic target.

## 4. Materials and Methods

### 4.1. Cell Culture

Cholangiocarcinoma (CCA) cell lines KKU213L5 and KKU100 were maintained in Dulbecco’s Modified Eagle Medium (Thermo Fisher Scientific, Waltham, MA, USA) supplemented with 10% FBS, 100 units/mL of penicillin, 100 μg/mL of streptomycin and 0.25 μg/mL of amphotericin B. Cells were grown in a humidified incubator at 37 °C with 5% CO_2_.

### 4.2. MTT Assay

CCA cells were seeded into a 96-well cell culture plate at a density of 5 × 10^3^ cells/well. Overnight incubation was carried out for cell adhesion before being replaced with a control medium or cisplatin/oxaliplatin-derived DAMPs. Cells were further incubated for 24 or 48 h. Cell culture medium was removed and replaced with MTT reagent, and incubated for 4 h. Formazan crystal was then dissolved by DMSO and measured for light absorbance at 540 nm using a microplate spectrophotometer. The experiment was performed in triplicate and represented as %proliferation compared to the control treatment.

### 4.3. Annexin V/PI Staining

KKU-213L5 cells were plated into 6-well cell culture plates at density 5 × 10^3^ cells/well. After cell adhesion, the cell culture medium was replaced with fresh medium containing cisplatin or oxaliplatin at IC50 concentrations and incubated for 24 h. Treated cells were washed with ice-cold PBS and collected by trypsinization. Cells were stained with the Muse^®^ Annexin V & Dead Cell Assay Kit, according to the manufacturer’s instruction, and analyzed using the Muse analyzer (Millipore-Sigma Aldrich, St. Louis, MO, USA). Experiments were performed in triplicate and results shown as representative values.

### 4.4. LDH Activity Measurement

The LDH activity was measured using a lactate dehydrogenase activity assay kit (Millipore-Sigma Aldrich, St. Louis, MO, USA). All the reagents and samples were prepared according to the instructions. Sample blank, standard and drug-derived conditioned media were mixed with working reagent and the kinetics of light absorbance were observed at 450 nm using a microplate spectrophotometer. Experiments were performed in triplicate and results shown as representative values.

### 4.5. Apoptosis Protein Measurement

Treated CCA cells were harvested and lysed with RIPA lysis buffer plus protease inhibitor cocktail (AMRESCO, Solon, OH, USA). Apoptotic markers were investigated using Bio-Plex Pro RBM Apoptosis Assays panel 3 (BioRad, Hercules, CA, USA); reagents were prepared according to the kit’s manufacturer. Samples were mixed with capture beads and incubated for 1 h; after washing, the detection antibody was added and the mixture incubated for 1 h. The streptavidin-PE was subsequently added, and the sample was incubated for another 30 min. Beaded samples were then washed and analyzed using Bio-Plex^®^ 200 (BioRad, Hercules, CA, USA). The experiments were performed in triplicate, and the results are shown as representative values.

### 4.6. Preparation of Chemo-Drug-Derived DAMPs

KKU-213L5 cells were plated in a cell culture dish at a density of 2.5 × 10^6^ cells/dish. Cells were then treated for 24 h with cisplatin and oxaliplatin at µM, respectively. The cell culture medium was removed and washed 3 times with PBS, and replaced with a fresh cell culture medium. Cells were incubated for another 24 h. DAMPs containing culture medium were collected and centrifuged at 10,000× *g* for 20 min, and cell-free supernatant was collected and used as chemo-drug-derived DAMPs.

### 4.7. In-Solution Protein Digestion

The proteomic analysis was performed at the Proteomics Services Center, Center for Research and Innovation, Faculty of Medical Technology, Mahidol University, Nakhon Pathom, Thailand. The proteins were cleaned in samples by using a clean-up kit (GE Healthcare, Chicago, IL, USA). Then, the protein pellets were dissolved in 8 M urea and the protein concentration was measured by Bradford’s method (Bio-Rad protein assay, Bio-Rad Laboratory, Hercules, CA, USA).

For each sample, 25 μg of protein was reduced by incubation at room temperature for 30 min in reduction buffer (100 mM dithiothreitol in 100 mM TEAB). Then, an alkylating buffer (100 mM iodoacetamide in 100 mM TEAB) was added and the sample was incubated at room temperature in the dark for 30 min. It was then incubated again with reduction buffer at room temperature for 15 min for quenching, and then digested for 16 h at 37 °C using Trypsin, Gold (mass spectrometry grade; Promega, Madison, WI, USA). The sample was dried in a CentriVap DNA Concentrator (Labconco Co., Kansas City, MO, USA), resuspended in 0.1% formic acid (FA) and cleaned up using a C18 Zip Tip. The cleaned peptide was then dried in the CentriVap and stored at −80 °C until further processing. Finally, the sample was resuspended in 0.1% formic acid and the peptide concentration was measured by a Nano Drop 1000 (Thermo Fisher Scientific, Bremen, Germany).

### 4.8. Label-Free Nano-LC-MS/MS Analysis

Peptides were analyzed on an LC-MS/MS system including a nano-liquid chromatograph (Dionex Ultimate 3000, RSLCnano System, Thermo Scientific) in combination with a CaptiveSpray source/quadrupole ion trap mass spectrometer (Model Q-ToF Compact, Bruker, Germany). One microgram of peptides was enriched by a nano trap column, 100 μm i.d. × 2 cm, Acclaim PepMap100 C18 5 μm, pore size 100 Å, and separated using a PepMap100 C18 3 μm 75 μm × 500 mm LC column. Elution was performed using a linear gradient of 2–95% Solvent B over 220 min at a flow rate of 300 nL/min and a column temperature of 60 °C. There were 2 mobile phases: (A) 0.1% FA in water, and (B) 0.08% FA in 80% acetonitrile. The loading pump solvent consisted of 0.05% TFA in 2% acetonitrile. A gradient of mobile phase B was used as follows: 2% for 5 min, ramped to 35% for 180 min, then ramped to 55% for 20 min, and ramped to 95% for 10 min, then ramped down to 2% for 1 sec and re-equilibrated for 5 min. The drying gas flow and temperature were 5 L/min and 150 °C, respectively, and the nebulizer gas pressure was 0.2 bars. The MS acquisition rate was 6 Hz, and positive ionization mode was used, with a survey scan mass range of *m*/*z* 150–2200. AutoMSn CID fragmentation experiments were performed at low (4 Hz) and high (16 Hz) mass spectral rates for the two most intense precursor ions using 3-s dynamic exclusion. Sodium formate was used for internal calibration and injected with a string pump.

### 4.9. MS Data Processing and Statistical Analysis

For the discovery phase, raw MS data were processed with MaxQuant software (version 1.6.2.10) coupled with its built-in search engine, Andromeda, for protein identification. The default setting, with the human database downloaded from www.uniprot.org (accessed on 29 April 2022), was set to 1 as the label-free approach. Parameter settings used the default values, with the exception of the following. Oxidation of methionine and acetylation of the N-terminus were set as variable modifications, whilst carbamidomethyl modification of cysteine was set as a fixed modification. Bruker Q-TOF was selected as the instrument type, with peptide tolerance for the first and main searches set as 0.5 and 0.25, respectively. Trypsin/P was set for the identification of peptides with a maximum of two missed cleavages. Raw data were blasted against the Homo sapiens UniProt database. The false discovery rate (FDR) was set at 1% of the protein level. The TOF MS/MS match tolerance was set at 0.5 Da with label-free quantification. The match between run options in the software was used for mass and retention time, recalibrating between runs. LFQ was carried out and imported into Perseus software (version 1.6.8.0) for differential expression statistical analysis. Student’s *t*-test was used for comparison between the groups, with a cut-off set at *p*-Value < 0.05.

### 4.10. Enzyme-Linked Immunosorbent Assay (ELISA)

Cisplatin- and oxaliplatin-derived conditioned media were further analyzed for DAMP content including HMGB1, HSP90 and S100A12 by using ELISA kits (ElabSciences, Houston, TX, USA). Reagents and samples were prepared according to instructions. An amount of 100 µL of standard, sample blank or chemo-drug-derived conditioned medium was added into each well, and then incubated with captured antibody and secondary antibody and developed to chromogens, which were finally measured for light absorbance using a microplate spectrophotometer. The experiments were performed in triplicate, and the results are shown as representative values.

### 4.11. ATP Measurement

ATP determination was carried out using an ATP assay kit (Millipore-Sigma Aldrich, St. Louis, MO, USA). All the reagents and samples were prepared according to instructions. Sample blank, standard and drug-derived conditioned media were mixed with working reagent and incubated for 30 min. Fluorescence intensity was measured at Ex535/EM587 nm. Experiments were performed in triplicate and results shown as representative values.

### 4.12. Calreticulin Measurement

Cisplatin- and oxaliplatin-treated cells on slide chamber culture plates were washed and fixed with ice-cold methanol for 10 min, and then permeabilized with 0.1% Triton X-100 for 15 min. After blocking with 0.1% tween, cells were stained with anti-CRT Alexa Fluor^®^ 488 (Abcam, Cambridge, UK) overnight. Cells were then counter-stained with DAPI and visualized under a fluorescent microscope.

For flow cytometer analysis, cisplatin- and oxaliplatin-treated cells were washed and stained with anti-CRT Alexa Fluor^®^ 488 (Abcam, Cambridge, UK) for 1 h. The expression of ecto-CRT was analyzed using the CytoFLEX cell sorter (Beckman Coulter, Brea, CA, USA).

### 4.13. Cell Migration Assay

Investigation of the migration ability of CCA cells was performed using the wound scratch assay. The KKU-213L5 and KKU100 (2.5 × 10^5^ cells/well) cells were plated in 6-well cell culture plates. A wound was scratched using a p200 pipette tip and washed three times with PBS solution. Control condition medium and chemo-drug-derived DAMP-conditioned medium were added to CCA cells. The wound size was observed and measured after 0 h, 12 h and 24 h under an inverted microscope. The wound area was analyzed using ImageJ software and represented as % migration.

### 4.14. Immunoblotting

CCA cells were incubated with chemo-drug-derived DAMPs at indicated times. Treated cells were washed with ice-cold PBS before cell lysis using RIPA lysis buffer plus protease inhibitor cocktail (AMRESCO, Solon, OH, USA). The protein was then separated by sodium dodecyl sulfate–polyacrylamide gel electrophoresis (SDS-PAGE) and transferred onto polyvinylidene fluoride (PVDF) membranes. After this, the non-specific binding was blocked with 5% skim milk buffer for 1 h before washing with TBST buffer. The membranes were then incubated with each primary antibody, anti-pAkt, anti-Akt, anti-pERK, anti-ERK, anti-Cyclin D1 and anti-β actin (Cell Signaling, Danvers, MA, USA) antibodies, with gentle shaking at 4 °C overnight. Then, membranes were washed with TBST and incubated with horseradish peroxidase (HRP)-linked anti-rabbit antibody (Cell Signaling, Danvers, MA, USA) for 1 h at room temperature, and washed again before being incubated with a detection reagent. The images were developed using Chimidoc™ XRS (Bio-rad, Hercules, CA, USA) and analyzed with Image Lab (Bio-rad, Hercules, CA, USA).

## 5. Conclusions

In this study, we first demonstrated the effect of cisplatin and oxaliplatin on DAMPs’ expression in CCA cells. Proteomic analysis and biological assays revealed that over forty DAMPs are released due to cell toxicity from chemotherapeutic agents. Interestingly, chemotherapeutic-derived DAMPs could promote cancer proliferation and migration on CCA cells through survival signaling pathways. This emphasizes that DAMPs should be considered for use as biomarkers to indicate tumor progression or recurrence after the completion of chemotherapy treatment. Increment of DAMPs in circulation should remind to tumor progression or recurrent according to its protumor activity.

## Figures and Tables

**Figure 1 ijms-23-10540-f001:**
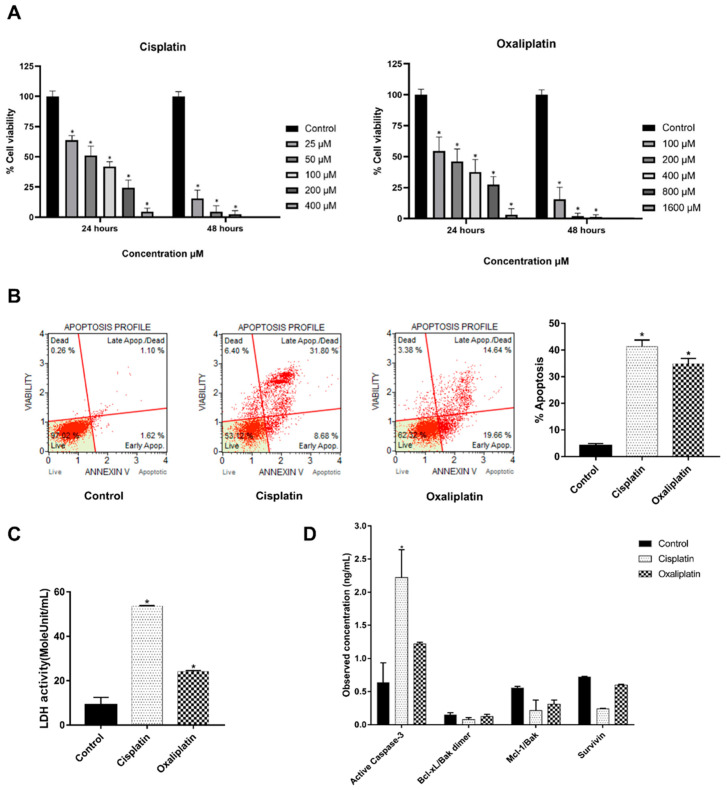
Cisplatin and oxaliplatin induced CCA cell toxicity. (**A**) KKU213L5 was incubated with chemo drugs as indicated concentrations and times. Cell viability as accessed by MTT method. (**B**) CCA cell was treated with drugs at IC50 concentration for 24 h before stained with annexin V/PI and analyzed by Muse analyzer. (**C**) Cell conditioned media were collected and measured LDH activity. (**D**) Treated cells were lysed and harvested intracellular protein for measure apoptotic marker by Bio-Plex Pro RBM Apoptosis Assays panel 3. * *p* < 0.05.

**Figure 2 ijms-23-10540-f002:**
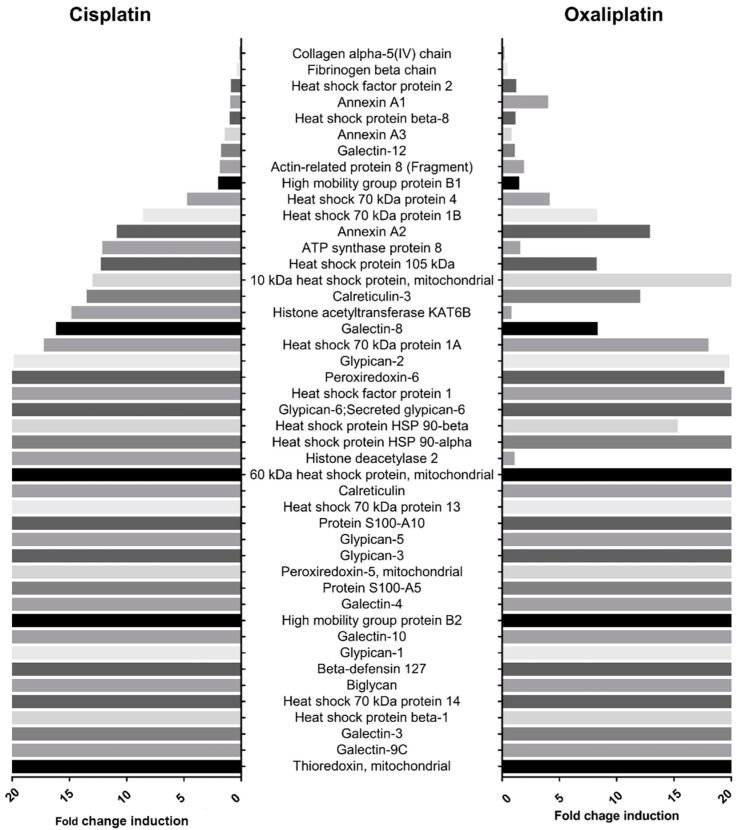
Representative DAMPs released from chemotherapy treatment. Overexpressed of 45 DAMPs were investigated by short-gun proteomic and plotted against fold induction, which compared cisplatin and oxaliplatin treatment.

**Figure 3 ijms-23-10540-f003:**
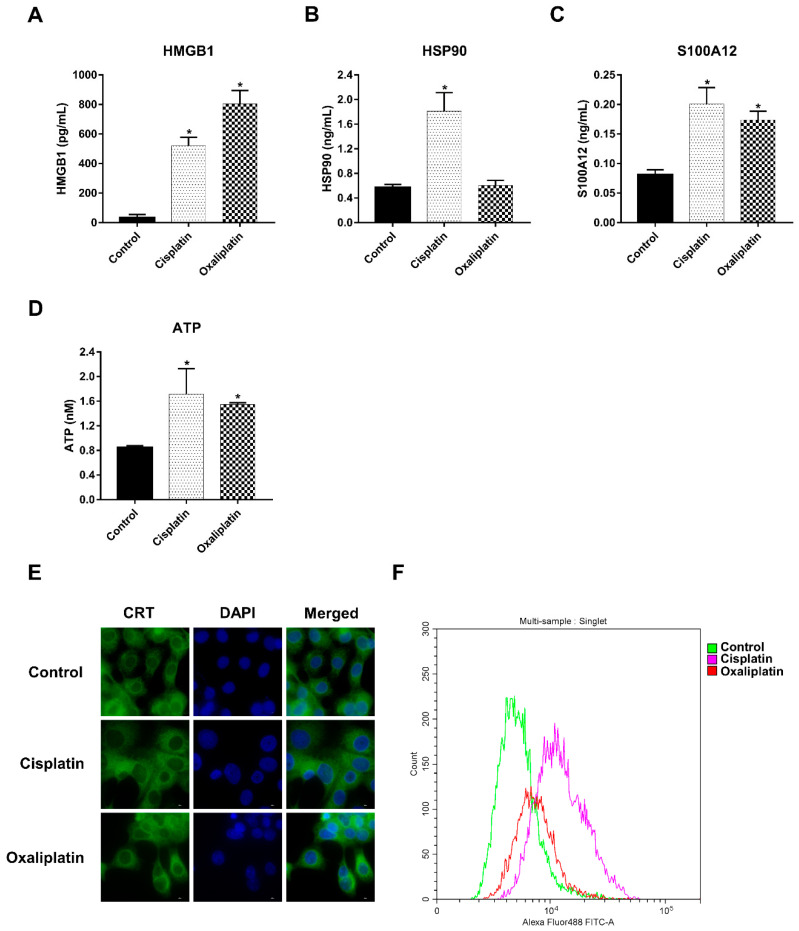
Cisplatin and oxaliplatin promoted DAMPs expression on CCA cell. The conditioned media of control or chemo drugs treatment were collected and measured the expression of (**A**) HMGB1, (**B**) HSP90 and (**C**) S100A12 by ELISA kit, and (**D**) ATP by fluorometric assay. Fluorescent staining (**E**) and flow cytometry cell surface expression (**F**) of CRT were investigated after treated with cisplatin and oxaliplatin. * *p* < 0.05.

**Figure 4 ijms-23-10540-f004:**
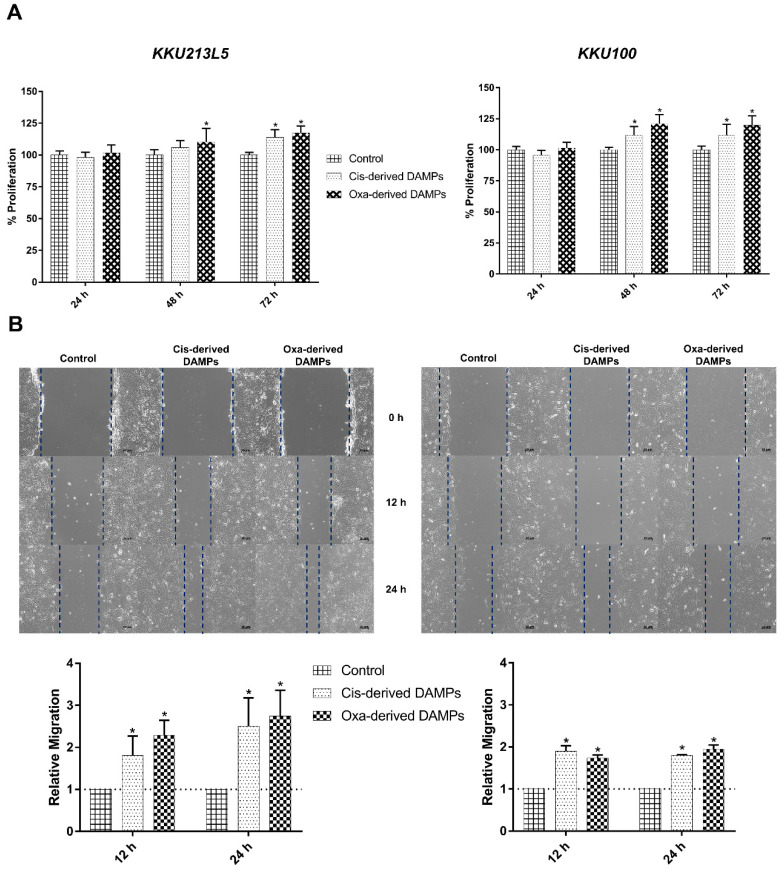
Chemo drugs-derived DAMPs promoted protumor activities. (**A**) KKU213L5 and KKU-100 were incubated with cisplatin- and oxaliplatin-derived conditioned medium and incubated for indicated times, cell viability was measured and quantified as %proliferation compared with control medium. In addition, the conditioned media were used to challenge with cell scratch assay (**B**). Wound closure was monitored at 12 and 24 h after treatment and measured wound area by ImageJ software version 1.8.0_172 (NY, USA). Data represented as relative migration that compare with control group. * *p* < 0.05.

**Figure 5 ijms-23-10540-f005:**
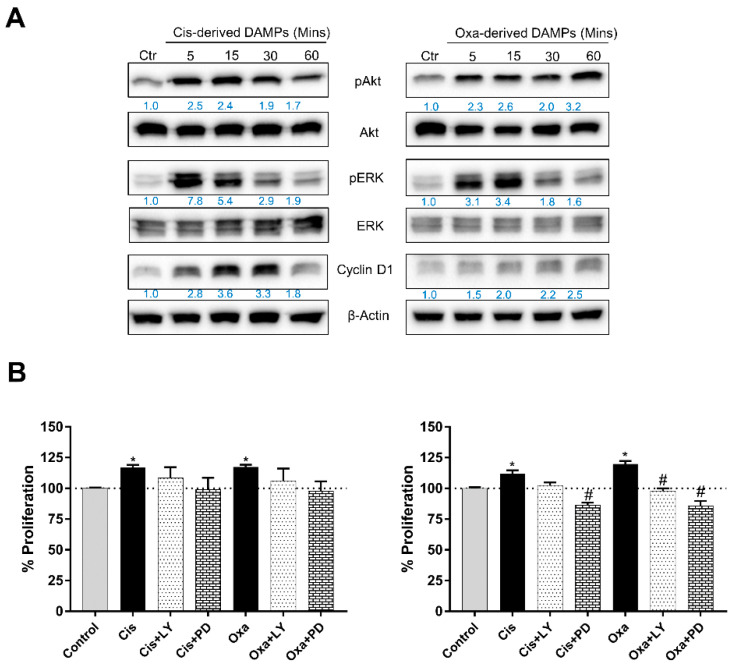
Cisplatin- and oxaliplatin-derived DAMPs induced CCA proliferation mediated pro-survival signaling. (**A**) KKU213L5 and KKU100 cells were incubated with conditioned media as indicated times, and survival associated signaling proteins were determined using immunoblotting. β-actin was used as loading control. (**B**) For phenotypic confirm, additional of 15 µM of LY294002 or PD98059 were rechallenged with proliferation activity of DAMPs’ conditioned media. Cell viability was measured and quantified as %proliferation. * *p* < 0.05 compared with control treatment, # *p* < 0.05 compared with cisplatin- and oxaliplatin-derived DAMPs treatments.

**Figure 6 ijms-23-10540-f006:**
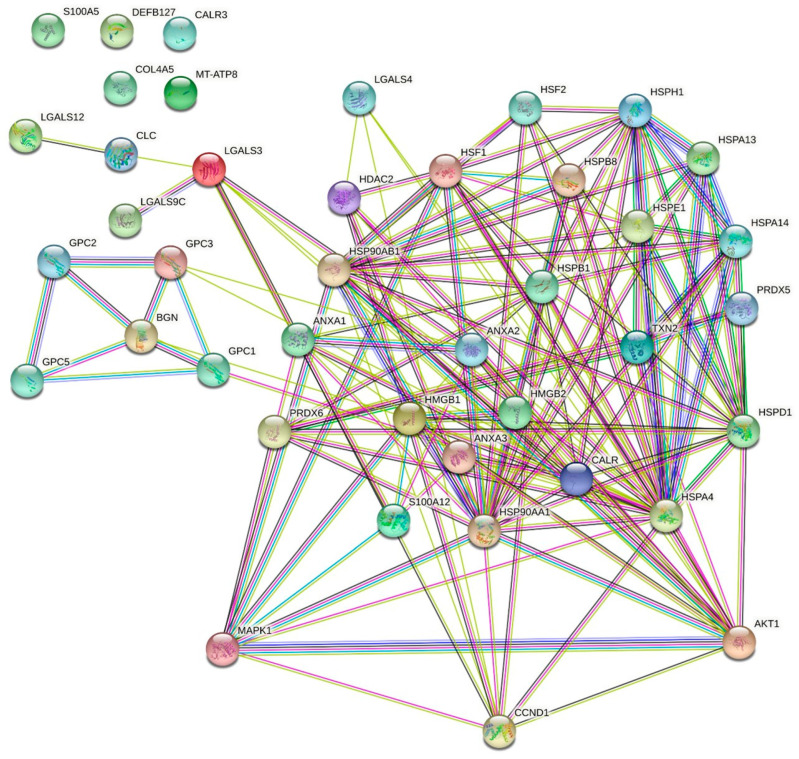
Postulate interaction of DAMPs and representative signaling proteins. Protein–protein interaction analysis was used to represent the association of DAMPs and crucial protumor signaling by using STRING software (http://string-db.org, accessed on 15 July 2022).

**Table 1 ijms-23-10540-t001:** Overexpression of DAMPs and DAMPs-related protein in cisplatin- and oxaliplatin-derived conditioned medium. Quantitative intensity of protein expression is mean of three assessments.

Accession No.	Protein Name	MW (kDa)	Quantitative Intensity			Fold-Change		*p*-Value	
			Control	CIS	OXA	CIS	OXA	CIS	OXA
P61604	10 kDa heat shock protein, mitochondrial	11	15,427	198,630	482,330	12.88	31.27	0.0009	0.0001
P10809	60 kDa heat shock protein, mitochondrial	61	298,183	10,427,833	6,603,367	34.97	22.15	0.0009	0.0035
P04083	Annexin A1	39	709,853	620,593	2,785,267	0.87	3.92	0.4866	0.0038
P07355	Annexin A2	39	159,363	1,715,067	2,038,033	10.76	12.79	0.0020	0.0067
P12429	Annexin A3	36	1,748,067	2,358,433	1,273,867	1.35	0.73	0.1356	0.0970
V9M9E1	ATP synthase protein 8	8	1435	17,277	2138	12.04	1.49	0.0015	0.2006
Q9H1M4	Beta-defensin 127	11	14,510	2,656,900	1,967,767	183.11	135.61	0.0014	0.0009
A6NLG9	Biglycan	35	11,992	2,858,133	2,728,933	238.34	227.57	0.0001	0.0004
P27797	Calreticulin	48	71,323	2,508,657	1,840,800	35.17	25.81	0.0850	0.0014
Q96L12	Calreticulin-3	45	123,947	1,659,800	1,481,167	13.39	11.95	0.0000	0.0007
Q05315	Galectin-10	16	2819	476,200	599,833	168.91	212.76	0.0004	0.0009
Q96DT0	Galectin-12	38	527,307	874,453	530,443	1.66	1.01	0.0100	0.9642
P17931	Galectin-3	26	40	286,617	451,240	7165.42	11,281	0.0001	0.0005
P56470	Galectin-4	36	5787	481,857	600,410	83.26	103.75	0.0002	0.0001
O00214	Galectin-8	36	82,826	1,330,867	681,423	16.07	8.23	0.0001	0.0012
Q6DKI2	Galectin-9C	40	40	641,557	474,763	16,038.92	11,869.08	0.0001	0.0031
A0A024R4D5	Glypican-1	54	10,264	1,750,100	2,619,967	170.51	255.25	0.0001	0.0029
A0A0J9YXG7	Glypican-2	14	88,987	1,760,067	1,756,433	19.78	19.74	0.0001	0.0019
Q8IYG2	Glypican-3	66	1764	84,857	118,769	48.09	67.31	0.0018	0.0007
A0A087WX91	Glypican-5	36	28,399	1,273,533	565,997	44.84	19.93	0.0004	0.0023
Q9Y625	Glypican-6; Secreted glypican-6	63	107,510	3,483,800	2,420,267	32.40	22.51	0.0003	0.0003
P48723	Heat shock 70 kDa protein 13	52	36,146	1,438,600	775,427	39.80	21.45	0.0005	0.0856
Q0VDF9	Heat shock 70 kDa protein 14	55	3834	1,665,500	2,451,500	434.40	639.41	0.0032	0.0006
P0DMV8	Heat shock 70 kDa protein 1A	70	70,951	1,216,900	1,271,600	17.15	17.92	0.0001	0.0006
P0DMV9	Heat shock 70 kDa protein 1B	70	361,193	3,071,400	2,968,700	8.50	8.22	0.0007	0.0032
P34932	Heat shock 70 kDa protein 4	94	941,070	4,376,200	3,807,100	4.65	4.05	0.0016	0.0032
Q00613	Heat shock factor protein 1	57	45,915	1,332,633	2,756,267	29.02	60.03	0.0010	0.0001
Q03933	Heat shock factor protein 2	60	1,828,433	1,462,167	2,063,067	0.80	1.13	0.2457	0.5166
Q92598	Heat shock protein 105 kDa	97	438,553	5,339,067	3,570,400	12.17	8.14	0.0008	0.0016
P04792	Heat shock protein beta-1	23	40	234,067	695,710	5851.67	17,392.75	0.0001	0.0008
Q9UJY1	Heat shock protein beta-8	22	16,375	15,190	17,364	0.93	1.06	0.6088	0.7492
P07900	Heat shock protein HSP 90-alpha	85	86,179	2,883,267	2,937,467	33.46	34.09	0.0029	0.0006
P08238	Heat shock protein HSP 90-beta	83	145,503	4,792,967	2,217,933	32.94	15.24	0.0003	0.0043
P09429	High mobility group protein B1	25	134,723	260,000	187,963	1.93	1.40	0.0120	0.0820
P26583	High mobility group protein B2	24	19,466	2,220,067	5,347,000	114.05	274.69	0.0026	0.0010
A0A3B3IU71	Histone acetyltransferase KAT6B (Fragment)	1	3951	58,169	2797	14.72	0.71	0.0001	0.0966
Q92769	Histone deacetylase 2	55	4963	171,967	4926	34.65	0.99	0.0001	0.9808
P30044	Peroxiredoxin-5, mitochondrial	22	4069	258,147	456,563	63.44	112.20	0.0011	0.0030
P30041	Peroxiredoxin-6	25	51,694	1,386,367	997,103	26.82	19.29	0.0008	0.0014
D3DV26	Protein S100-A10 (Fragment)	22	81,311	3,423,500	4,124,333	42.10	50.72	0.0063	0.0008
P33763	Protein S100-A5	11	16,688	1,242,400	938,803	74.45	56.26	0.0004	0.0002
Q99757	Thioredoxin, mitochondrial	18	40	1,932,133	2,071,300	48,303.33	51,782.50	0.0048	0.0010

## Data Availability

The data presented in this study are available on request from the corresponding author.

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
