# Peer review of "Tumor-Promoting Activity and Proteomic Profiling of Cisplatin/Oxaliplatin-Derived DAMPs in Cholangiocarcinoma Cells"

_ijms, 2022, doi:10.3390/ijms231810540_

Round 1
Reviewer 1 Report
Before assuming this manuscript for publication, there are minor revisions to be considered.
Please, could you expand on and strengthen the conclusions?
In particular, the hypothesis of use of DAMPs "as biomarkers to indicate tumor progression or recurrence after the completion of chemotherapy treatment".
Please, revise the manuscript on the basis of these suggestions (as reported in attached pdf):
- page 2 line 72: “LDH”, explain the acronym of lactate dehydrogenase
- page 3, line 78: "mediums"; the correct plural of medium is “media”, as the authors reported in Materials and Methods
- page 7 line 118: as reported for line 78
- page 7 line 124: the terms “in vitro” should be in italic
- page 8 line 135: as reported for line 78
- page 9 line 159: as reported for line 78
- page 9 line 161: as reported for line 78
- page 10 line 177: the acronym of “immunogenic cell death”, ICD has already been introduced in the Abstract and in the Introduction
- page 11 line 207: the term “ex vivo” should be in italics
- page 11 line 235: at 37, put the symbol of temperature and at CO2, put the number subscript
- page 12 line 237: at 5x103, put the number 3 superscript
- page 12 line 245: at 5x103, as reported for line 237
- page 12 line 270: at 2.5x106, put the number 6 superscript
- page 14 line 361: %migration, put one space
- In Table 1, the columns and/or numbers must be arranged in such a way as to be able to read the numbers in their entirety; idem for the title of the columns.

Reviewer 2 Report
The manuscript is well written and the authors did a commendable job in performing all the experiments meticulously. The manuscript mainly describes the drug-induced 19 DAMPs’ expression and their role in tumor progression after treatment with well-known chemotherapeutic drugs, cisplatin, and oxaliplatin. The general concepts behind this paper and its conclusions are relatively straightforward. There are articles published before with the same concept but in different cancer (https://doi.org/10.1016/j.oraloncology.2019.06.016). There are some areas that, if clarified, will make this a much more engaging article. These points are:
1. A study of the cellular location of the DAMPs would help the readers understand the process. Especially, when the DAMPs are released from the nucleosome (or any source) and move to the cytoplasm or outside the cell.
2. Is there any changes taking place on the cell surface of the cells treated with the DAMPs containing media?
3. “To evaluate the biological activity of chemo-drug-derived DAMPs, a conditioned medium of cisplatin- and oxaliplatin-treated CCA cells were inspected in terms of in vitro cancer proliferation and migration.” How and through which channels are secretory DAMPs going inside the untreated cells?
4. DAMPs were initially considered to be exclusively released from necrotic cells. Please explain how the DAMPs in your study were from apoptotic cells, not necrotic cells.
Round 2
Reviewer 2 Report
Authors explained all the concerns diligently.